# Gathered Wild Food Plants among Diverse Religious Groups in Jhelum District, Punjab, Pakistan

**DOI:** 10.3390/foods10030594

**Published:** 2021-03-11

**Authors:** Muhammad Majeed, Khizar Hayat Bhatti, Andrea Pieroni, Renata Sõukand, Rainer W. Bussmann, Arshad Mahmood Khan, Sunbal Khalil Chaudhari, Muhammad Abdul Aziz, Muhammad Shoaib Amjad

**Affiliations:** 1Department of Botany, Hafiz Hayat Campus, University of Gujrat, Gujrat, Punjab 50700, Pakistan; majeedmohal@gmail.com (M.M.); khizar.hayat@uog.edu.pk (K.H.B.); 2University of Gastronomic Sciences, Piazza Vittorio Emanuele II 9, 12042 Pollenzo/Bra (Cuneo), Italy; a.pieroni@unisg.it (A.P.); m.aziz@studenti.unisg.it (M.A.A.); 3Department of Medical Analysis, Tishk International University, Erbil 4401, Iraq; 4Department of Environmental Sciences, Informatics and Statistics, Ca’ Foscari University of Venice, Via Torino 155, 30172 Mestre, Italy; renata.soukand@unive.it; 5Department of Ethnobotany, Institute of Botany, Ilia State University, Tbilisi 0162, Georgia; rbussmann@gmail.com; 6Department of Botany, Govt. Hashmat Ali Islamia Degree College Rawalpindi, Rawalpindi 46000, Pakistan; arshadbotanist@gmail.com; 7Department of Botany, Sargodha Campus, Institute of Molecular Biology and Biotechnology, The University of Lahore, Sargodha 40100, Pakistan; sunbal.khalil@imbb.uol.edu.pk; 8Department of Botany, Women University of Azad Jammu and Kashmir, Bagh 12500, Pakistan

**Keywords:** ethnobotany, wild food plants, traditional food, religious diversity, bio-cultural heritage, local resources

## Abstract

Recent ethnobotanical studies have raised the hypothesis that religious affiliation can, in certain circumstances, influence the evolution of the use of wild food plants, given that it shapes kinship relations and vertical transmission of traditional/local environmental knowledge. The local population living in Jhelum District, Punjab, Pakistan comprises very diverse religious and linguistic groups. A field study about the uses of wild food plants was conducted in the district. This field survey included 120 semi-structured interviews in 27 villages, focusing on six religious groups (Sunni and Shia Muslims, Christians, Hindus, Sikhs, and Ahmadis). We documented a total of 77 wild food plants and one mushroom species which were used by the local population mainly as cooked vegetables and raw snacks. The cross-religious comparison among six groups showed a high homogeneity of use among two Muslim groups (Shias and Sunnis), while the other four religious groups showed less extensive, yet diverse uses, staying within the variety of taxa used by Islamic groups. No specific plant cultural markers (i.e., plants gathered only by one community) could be identified, although there were a limited number of group-specific uses of the shared plants. Moreover, the field study showed erosion of the knowledge among the non-Muslim groups, which were more engaged in urban occupations and possibly underwent stronger cultural adaption to a modern lifestyle. The recorded traditional knowledge could be used to guide future development programs aimed at fostering food security and the valorization of the local bio-cultural heritage.

## 1. Introduction

Wild food plants have remained an important ingredient of traditional food basket systems especially in remote communities around the globe [1]. However, due to dramatic socio-cultural shifts local communities are facing and global climate change, dependence on wild food plants has drastically decreased in many areas. Food industrialization and globalization have severely impacted traditional food systems, especially in rural communities [2]. Consequently, traditional/local environmental knowledge (TEK) linked to wild food plants is becoming more and more endangered, and in some places of the world, it has already disappeared [3]. In recent decades, scientists have recorded several complex TEK systems associated to wild food plants, especially in marginalized areas. However, very few ethnobotanical field studies have focused on the cross-cultural and cross-regional comparison of TEK associated to wild food plants, despite the fact that cultural diversity shapes TEK [4,5,6,7,8,9].

In many regions of the world, inhabitants of rural areas depend on wild plants as food [10] and a large number of wild plant species occurring in a great variety of habitats are consumed [11,12]. Recent works have addressed the role that religious affiliation may play in shaping folk wild food plant uses and cuisines, since this factor shapes in many areas of the world kinship relations and the vertical transmission of plant and gastronomic knowledge [13,14,15]. However, all over the world wild plants have been more frequently consumed in the past [10]. There are over 20,000 species of wild edible plants in the world, yet fewer than 20 cultivated species now provide 90% of our main staples [16].

The collection and culinary use of wild plants for food are part of the bio-cultural heritage of local communities and therefore can foster their future sustainability [17,18]. During the last decades, a large number of publications have documented the ethnobotany of wild food plants, but only sporadically scholars have tried to articulate the evaluation of socio-cultural and economic factors possibly influencing foraging [19,20,21,22,23,24,25,26,27,28,29,30,31,32]; simultaneously, research on specific domains of the plant foodscape, such as that of fermentation of local plants (sometimes wild) is exponentially growing [33,34,35,36,37,38,39,40,41,42,43,44,45].

Pakistan comprises remarkable natural resources, and a large variety of religious faiths and linguistic communities using a wide range of wild food plants [46]. Many rural communities in Pakistan live closely attached to their natural resources [47] and wild food plants are often consumed for food [48]. A few comparative studies have very recently addressed the cross-cultural dimension of wild food plants gathering and use in Pakistan, and highlighted the role of diverse linguistic and religious groups [49,50,51].

In order to further evaluate this trajectory, the current study focused on six religious groups (Sunni and Shia Muslims, Christians, Hindus, Sikhs, and Ahmadis—also named Qadiani in official Pakistani documents, despite this term is considered sometime derogatory by the community), speaking eleven different languages (Urdu, Punjabi, Phtohari, Gojri, Pahari, Hindko, Saraiki, Sindhi, Pashto, Kashmiri, and Hindi) in Jhelum District, Punjab, NE Pakistan. 

The main aim of our research was to record local knowledge related to wild food plants and also to provide baseline documentation to help local stakeholders revitalizing their food traditions. We particularly explored the impact of religious and linguistic affiliation on the gathering, utilization and consumption of wild food plants in 27 villages in Jhelum district, Punjab, Pakistan, hypothesizing that there could be some differences between different faiths.

The specific research objectives of this study were:to explore and record the diversity of wild food plants gathered in Jhelum,to evaluate the local food and possible traditional perceptions,to compare the mentioned wild food plants among the six selected religious faith groups in order to possibly understand cross-cultural similarities and differences and to better understand the cultural context supporting the use of wild food plants found in Jhelum district.

## 2. Materials and Methods

### 2.1. Study Area

The study area of Jhelum district is located North of the river Jhelum and is bordered by Rawalpindi district in the North, Sargodha and Gujrat districts in the South, Azad Jammu and Kashmir in the East, and Chakwal district in the West [52,53]. The population of the district is 1.22 million, and 71% of the population lives in rural areas, while the remaining 29% are urban [54]. The climatic conditions are semi-arid, warm-subtropical, characterized by warm summers and severe winters. Jhelum is a semi-mountainous area (Figure 1), with a mean annual rainfall of 880 mm. The annual average temperature reaches 23.6 °C. Jhelum is home to the world’s second largest salt mine (Khewra) covering about 1000 ha [53,55]. The people of Jhelum have a diverse culture with distinct modes of life, traditions, and beliefs [56]. The ethnic groups of the area show a strong connection to wild plants which often have cultural and medicinal significance [57].

The study was conducted in 27 remote villages (Figure 2), all of which contained rivers, mountains, forests, salt mines, and valleys. Some typical and important attributes including landscapes, vegetation, geology and soil, and rangeland are shown in Figure 1.

### 2.2. Field Study

The ethnobotanical field research was conducted from March to November 2020. Study participants were selected through snowball sampling focusing on middle-aged and elderly inhabitants (range: 40–90 years old), especially farmers, herders, and housewives. Selected interviewees belonged to different religious faiths and different language groups. Twenty participants (men and women) from each religious group were selected and participated in the survey. The characteristics of the study participants from the 27 visited villages and their different socio-cultural and economic attributes are reported in Table 1. 

Prior to starting an interview, oral informed consent was obtained, and the Code of Ethics of the International Society of Ethnobiology [58] was followed. Semi-structured interviews were conducted in the national language, Urdu, and some local languages (Punjabi, Saraiki, Pothohari, Gojri, Hinko, Pahari, Kashmiri, Sindhi, and Hindi) with the help of translators. The information collected focused on the gathering and consumption patterns of wild plants as cooked vegetables, raw snacks, salads, herbal drinks, recreational herbal teas, jams, and for fermentation following Kujawska and Łuczaj [59]. Particular questions were focused on the use of wild plants in daily food habits or in food fermentation, and the consumption of edible wild food plants [49]. Local names of collected taxa were recorded in eleven different local languages.

During the interviews qualitative ethnographic data was documented following Termote et al. [60]. The recorded wild food plants were collected from the study area and were identified using the Flora of Pakistan [61,62,63]. After correct identification, each taxon was given a voucher specimen number and deposited in the Herbarium of the Department of Botany, University of Gujrat, Punjab, Pakistan. For nomenclature, the Plant List database [64] was followed for plants, and the Index Fungorum [65] for the single recorded mushroom taxon. The plant family nomenclature follows the Angiosperm Phylogeny Group [66].

### 2.3. Data Analysis

The documented data was stored in two main binary data spreadsheets (1. Species gathered for any use; 2. Species gathered for specific use) across the six local religious communities and compared through Venn diagrams and pairwise Jaccard’s dissimilarity using the R Statistical Package [67,68,69].

The Jaccard Index (JI) was calculated as:J(X, Y) = |X∩Y|/|X∪Y|

X = Individual set of plant usages documented by group X

Y = Individual set of plant usages documented by group Y

By using JI, Jaccard’s distance (JD) was calculated as:D(X,Y) = 1 − J(X,Y)

Moreover, a qualitative comparison with other studies on wild food plants carried out in Pakistan [49,50,51,52,70,71,72] was conducted.

## 3. Results and Discussion

### 3.1. Reported Wild Food Plants and Their Uses 

A total of seventy-eight taxa (77 vascular plants and one mushroom) were gathered and consumed in different ways in the study area (Table 2). The most commonly used wild food plant species were native, with the exception of *Agave americana*, *Amaranthus spinosus*, *Sonchus oleraceus*, *Tephrosia purpurea*, *Trigonella corniculata*, *Salvia moorcroftiana*, *Salvia nubicola*, *Solanum incanum*, *Chenopodium album*, and *Portulaca quadrifida*, which were grown as herbs or grew wild as weeds in anthropogenically disturbed locations. A total of nine different typologies of food preparations were identified: chutneys (a family of spicy condiments and sauces prototypical of South Asian cuisines); cooked vegetables; fermented preparations; herbal drinks (plant material infused in cold water); herbal teas (plant material infused in hot water); jams; raw snacks (consumed singly, mostly in the field at the collection site); salads (raw plants consumed at the dining table as a starter and/or in conjunction with other food items); and seasoning/spices.

The most commonly quoted wild food plants and the typologies of their food preparations are reported in Figure 3.

The most important site for the gathering of wild food plants were grasslands, found sometimes at high elevations, where people normally bring animals for grazing. Summer herders were the most knowledgeable ethnobotanical informants and this show the importance of the link between resilience of wild food plant knowledge and the survival of pastoralist activities. However, the transmission of ethnobotanical practices from elders to the younger generation is continuously decreasing due to the generation gap and fast changing lifestyle. With the modernization of life, the younger generation is moving towards cities for education and business opportunities, which is one of the major reasons for the decline of TEK described in many ethnobotanical studies.

Some important wild food plants (Figure 4) and dishes prepared by the visited communities were available for photographing (Figure 5). Traditional culinary processing included cooking the plants as vegetables (43 mentions), followed by raw snacks (33), confirming what documented in other ethnobotanical studies too [73,74,75]. Raw snacks were eaten especially by transhumant herders, and it has been shown that herding develops specific linkages between humans and their surrounding ecosystem [76,77,78,79]. Herding is also linked to the use of particular types of wild food plants: for example, in Iraq and Kurdistan shepherds consumed more raw snacks than nearby horticulturists [9,76]. Moreover, pastures have been documented as very important gathering habitats of wild food plants [80,81].

Leaves were the most used plant part (38 times used), especially in salads, herbal teas, herbal drinks, as raw snacks, in chutneys, and as cooked vegetables. One third of the reported plants (27 taxa) were only gathered during the spring season.

It was noted that sweet fruits in particular were consumed as raw snacks especially by local communities with a herding lifestyle. Thirty wild food plants were consumed as raw snacks by all religious faith groups, especially *Capparis decidua*, *Caragana ambigua, Cucumis melo*, *Lathyrus aphaca*, *Lathyrus sativus*, *Phoenix sylvestris*, *Salvadora persica*, *Solanum americanum*, *Solanum incanum*, *Solanum villosum*, *Ziziphus jujube*, *Ziziphus nummularia*, *Ziziphus oxyphylla*, and *Ziziphus spina-christi*, many also earlier reported by Sõukand and Kalle [82]. Although *Solanum americanum* was recognized as containing toxic alkaloids [83], especially in its fruit [84], informants used fruits as raw snacks without reporting any toxic effects. Similarly, some other important food preparations in the study area were herbal drinks, salads and chutney (Figure 5).

On a global scale, it has been found that folk knowledge has been decreasing, mostly due to modern lifestyle changes and urbanization [50,71,85,86,87,88,89,90]. Gathering wild food plants is linked to local biodiversity and especially local cultural practices [91] and in our field study wild plant knowledge among younger informants was limited, similar to what was found in many other studies, for example, Kalle and Sõukand [92].

### 3.2. Cross-Religious Comparison

Cross-religious comparison of the used wild food plants (Figure 6) shows a remarkable homogeneity and the absence of any plant cultural markers (i.e., plants used by one group only); at the same time, however, not a single taxa is used by all the six considered groups and the majority of recorded wild food plants are used by three to four groups.

However, the Jaccard’s distance heat map (Figure 7) shows high dissimilarity between some groups. While both Muslim groups, Shias and Sunnis, appeared to be closest in their selection of the wild food plants, Hindus and Christians are the most distant. 

The heat map (Figure 7) allows us to distinguish two easily comparable clusters within the six religious groups: a subgroup of Shias, Sunnis, and Sikhs, which used the highest number of plants (from 56 to 73) and a subgroup using far fewer taxa (Christians, Hindus, and Ahmadis (from 34 to 47). Bearing in mind that Shias and Sunnis together used all 78 listed taxa, there is a clear pattern of dissimilarity among the second subgroup (Figure 8).

Sunnis, using a slightly higher number of taxa than Shias, had more similarities with all the other groups. This could be due to the fact that the Sunni faith is the dominant one in the study area.

Figure 9 shows the comparison among the six groups in terms of specific food uses of the recorded wild food plants; the diagram shows a high diversity as well as a few specific cultural markers.

The similarity heat maps on the typology of wild plants food uses (Figure 10) demonstrates similar tendencies, outlining even greater differences between Christians and Ahmadis compared to Hindus, and also showing more divergences even among Sunnis and Shias. This suggests that there is a higher similarity in the used wild food plants than the way taxa are actually consumed in the study area; moreover, each considered group retains unique wild food plant utilizations.

While our results show remarkable social and cultural exchanges between the different religious groups (sharing the same repertoire of plants), we can also see clear differences among the ways local food plants are actually used. This may to be linked to the different exposure the diverse religious groups have to traditional rural lifestyles and to nature. Nowadays, only Shias, Sunnis and Ahmadis have for example retained traditional livelihood practices (farming), while the local community members belonging to the three other faiths (Christians, Hindus, and Sikhs) are partially employed in city jobs, and some of them even practice as professional herbalists. The different relationships to farming that shape the differences in wild food plants-centered TEK among the groups may also be due to diverse levels of land access and land ownership. 

While members of the different religions in the study area generally do not intermarry, they very regularly interact in urban settings and this, over centuries, may have contributed to a homogenization of TEK and cultural adaptation to the dominant groups.

The study participants confirmed that the use of wild plant species as daily food has significantly decreased, as well as the use of wild food plants on special occasions and religious festivities. This may be due to the fact that study participants perceive nowadays foraging (collecting wild food plants) as very time consuming, while cultivated plants are relatively easy to purchase in the immediate vicinity, and especially in bulk if and when required on special occasions. These trends may further lead to rapid TEK erosion in the near future, and further ethnobotanical works documenting local uses of wild food plants could be crucial for the food security and the preservation of the bio-cultural heritage of rural communities [93,94].

Food taboos restricting the consumption of some plants and fruits under certain conditions have been described from many regions of the world, involving followers of various religions including Hindus [95]. Similarly, in this study, some Hindus participants reported that the fruits of *Ziziphus oxyphylla* and *Ziziphus jujuba* were gathered only in mountain areas, hilly slopes and scrubland in time of need as famine foods only. Hence, the Hindus, but others possibly as well, follow the specific rules in what they consume, especially like when pregnant or menstruating. Food taboos might influence the uses of certain wild plants with regard to seasons or a consumer’s health condition, gender or age [95]. The participants pointed out a few other idiosyncratic food uses of wild plants within specific groups as well; these uses mostly included *medicinal foods*, i.e., food preparations considered consumed for counteracting specific diseases or health conditions, or ritual uses linked to specific cultural beliefs. For example, the gum of *Acacia modesta* and *Acacia nilotica* is added in “halwa” (a local sweet prepared by using clarified butter and wheat bran), and recommended to women after childbirth to avoid general weakness and back pains among the Muslim participants. Similarly, Sikhs conveyed that a limited dosage (about 250–300 mL) of a herbal drink made with *Cannabis sativa* can induce activeness; the informants claimed that their ancestors use the same preparation during battles in the 19th and 20th century. The herbal tea prepared by using fruits of *Tribulus terrestris* is drunk by Hindu women in order to improve lactation. Finally, Muslims add leaves of *Ziziphus jujuba* and *Ziziphus numularia* in boiling water, and use them for bathing dead persons, as they perceive that would delay their decomposition until burial. The rest of documented wild edible plant species as food in this study may applicable to all gender, religion and age groups equally, and no associated food taboo is mentioned by any participant.

### 3.3. Comparison with the Pakistani Food Ethnobotanical Literature 

The comprehensive comparison with the Pakistani wild food ethnobotanical literature [49,50,51,52,70,71,72] of Pakistan showed that a remarkable number of species were documented as wild food plants, for the first time, in the study regions: *Acacia modesta, Acacia nilotica, Agave americana, Boerhavia repens, Capparis decidua, Chenopodium murale, Chenopodium vulvaria, Coprinus comatus, Corchorus depressus, Corchorus tridens, Cucumis melo, Dysphania ambrosioides, Fagonia indica, Gisekia pharnaceoides, Indigofera hochstetteri, Lathyrus sativus, Lepidium apetalum, Mentha arvensis, Mentha pulegium, Olea europaea, Phoenix sylvestris, Pistia stratiotes, Prosopis cineraria, Prosopis juliflora, Rhynchosia minima, Salvadora oleoides, Salvadora persica, Senna italica, Senna occidentalis, Solanum incanum, Tephrosia purpurea, Tribulus terrestris, Trigonella anguina, Trigonella corniculata,* and *Withania coagulans*.

Despite the fact that in our study three quarters of the wild food plants were reported also in other areas of northern Pakistan [50,52], pairwise Jaccard’s distance between our findings and those arising from field studies recently conducted in various regions of Pakistan shows little similarity and a very large diversification of wild food plant uses within the country (Figure 11). This may be explained by the very diverse geography and natural environments, as well as a remarkable cultural diversity, which ultimately and most importantly affect the diversity of food customs of the country.

Based on a comprehensive literature review, we found that some wild food plant species recorded in the current study have rarely been documented as food ingredients elsewhere in Pakistan and its neighboring countries. These include *Aerva javanica*, *Agave americana*, *Amaranthus spinosus*, *Boerhavia repens*, *Caragana ambigua*, *Commelina benghalensis*, *Convolvulus arvensis*, *Corchorus depressus*, *Corchorus tridens*, *Gisekia pharnaceoides*, *Indigofera hochstetteri*, *Lathyrus aphaca*, *Lepidium apetalum*, *Mentha royleana*, *Opuntia dillenii*, *Oxalis corniculata*, *Physalis divaricata*, *Pistia stratiotes*, *Polygonum plebeium*, *Rhynchosia minima*, *Trigonella anguina*, *Trigonella corniculata*, and *Veronica anagallis-aquatica.*

## 4. Conclusions

Our study reported seventy-seven plant taxa and one mushroom used as cultural foods among six different religions. The cross-religious comparison showed high overlap in the used taxa between Shias and Sunnis, who together used all listed taxa in the study region and contributed the most detailed information about specific, commonly used wild food plants. Comparison of the other four religious groups showed much less overlap between the groups and greater variation in the numbers of used plants. Urban Hindus and Christians used the least number of plants, followed by rural Sikhs and urban Ahmadis. A comparative analysis with the wild food plant literature of Pakistan showed a high diversification of wild plant uses in the study region, due to both environmental and cultural factors. This study also concluded that there is relatively higher homogeneity in use of plant species as food compared to method (preparations) of use of the same among the religious groups. Therefore, if one religious group prepares herbal drink of a plant species, the other might prefer to prepare jam of the same, depicting possession of unique recipes.

The inherited cultural knowledge of wild food plants of Hindus, Sikhs, Christians, and Ahmadis, in particular, faces the greatest challenges, as these groups have apparently undergone cultural adaptation to an urban, “modern” lifestyle. The present study may provide a foundation for the promotion of eco-tourism and for supporting sustainable development programs. Several of the recorded wild food plants are still sold in local markets (e.g., *Capparis*, *Mentha*, *Olea*, *Phoenix*, *Rhynchosia*, *Salvadora*, *Salvia*, *Senna*, *Solanum*, *Trigonella*, *Vicia*, and *Ziziphus* spp.) and this could represent the basis of wild food plant-centered local projects, aiming to revitalize TEK and generate small-scale economies providing some cash-income for rural communities.

## Figures and Tables

**Figure 1 foods-10-00594-f001:**
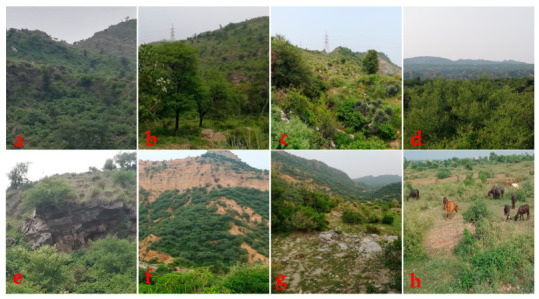
Diverse landscapes of Jhelum study area, NE Pakistan; (**a**–**d**): landscape depicting leading plant species associations (indicator species: *Acacia modesta, Acacia nilotica, Prosopis juliflora, Ziziphus numularia, Justicia adhatoda* and *Dodonea viscosa*); (**e**–**g**): exposed sedimentary bedrock stratification (age: Pre-Cambrian to Pliocene; composition: limestone, sandstone, shale, and dolomite) and sandy loam textured soil; (**h**): rangeland for livestock grazing.

**Figure 2 foods-10-00594-f002:**
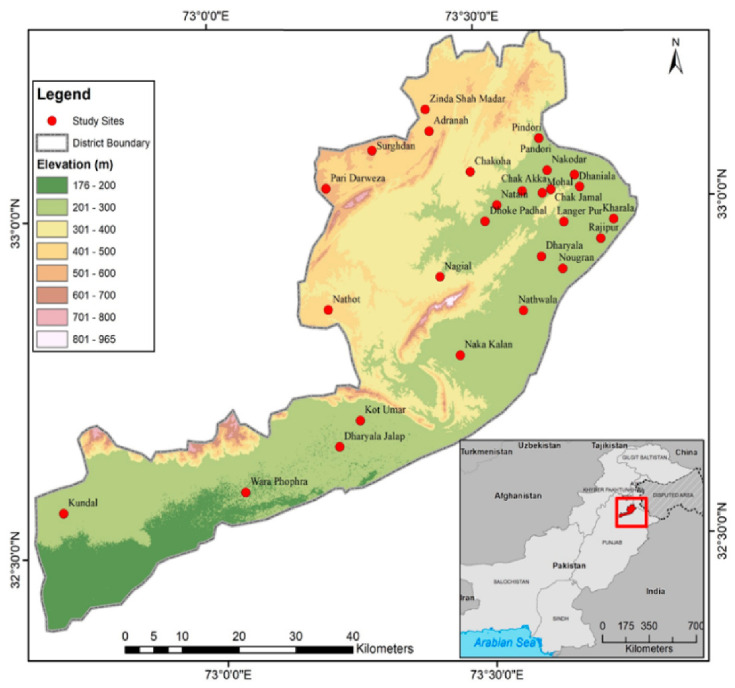
The map of the study area displaying the studied sites/village locations.

**Figure 3 foods-10-00594-f003:**
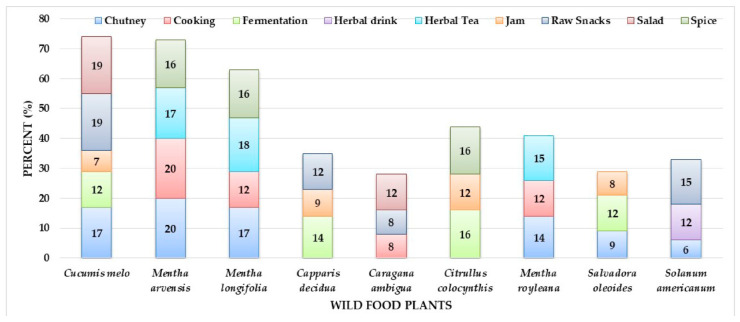
Most commonly used wild food plants and their uses.

**Figure 4 foods-10-00594-f004:**
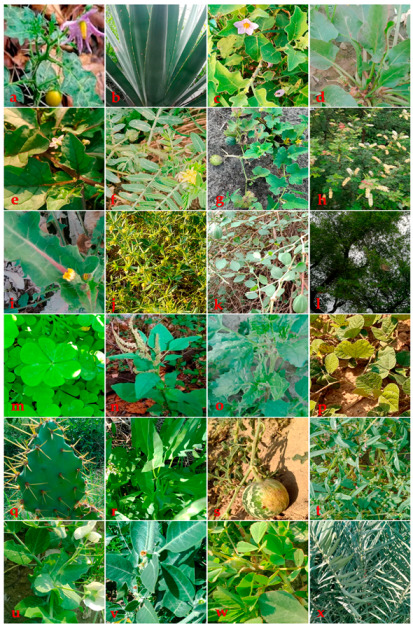
Some examples of wild food plants of Jhelum district: (**a**) *Solanum surattense*; (**b**) *Agave americana*; (**c**) *Solanum incanum*; (**d**) *Rumex dentatus*; (**e**) *Solanum americanum*; (**f**) *Tribulus terrestris*; (**g**) *Cucumis melo*; (**h**) *Acacia modesta*; (**i**) *Sonchus asper*; (**j**) *Fagonia indica*; (**k**) *Capparis decidua*; (**l**) *Ziziphus jujuba*; (**m**) *Oxalis corniculata*; (**n**) *Amaranthus spinosus*; (**o**) *Chenopodium murale*; (**p**) *Rhynchosia minima*; (**q**) *Opuntia dillenii*; (**r**) *Convolvulus arvensis*; (**s**) *Citrullus colocynthis*; (**t**) *Gisekia pharnaceoides*; (**u**) *Lathyrus sativus*; (**v**) *Withania coagulans*; (**w**) *Trigonella corniculata*; (**x**) *Phoenix sylvestris*.

**Figure 5 foods-10-00594-f005:**
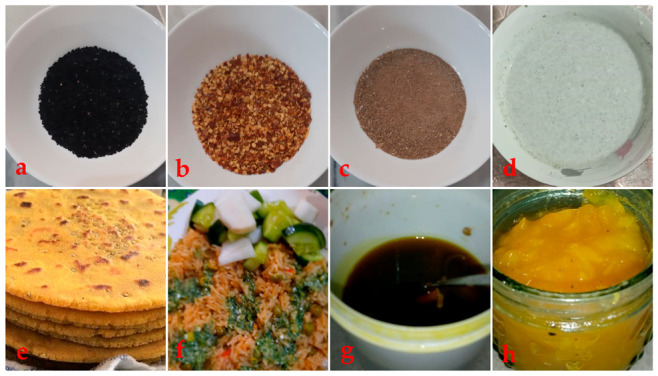
Traditional culinary uses of wild food plants by different linguistic and religious communities reported from the study area: (**a**) mixture of black pepper and *Mentha royleana*; (**b**) mixture of chilies and *Mentha pulegium*; (**c**) powdered *Citrullus colocynthis*; (**d**) powedered *Mentha arvensis* in yogurt; (**e**) bread made with rice flour with *Opuntia dillenii* pulp; (**f**) rice cooked with *Amaranthus viridis* seeds and *Cucumis melo* as salad; (**g**) herbal drink made with *Cannabis sativa*; (**h**) jam made by *Prosopis cineraria* fruits.

**Figure 6 foods-10-00594-f006:**
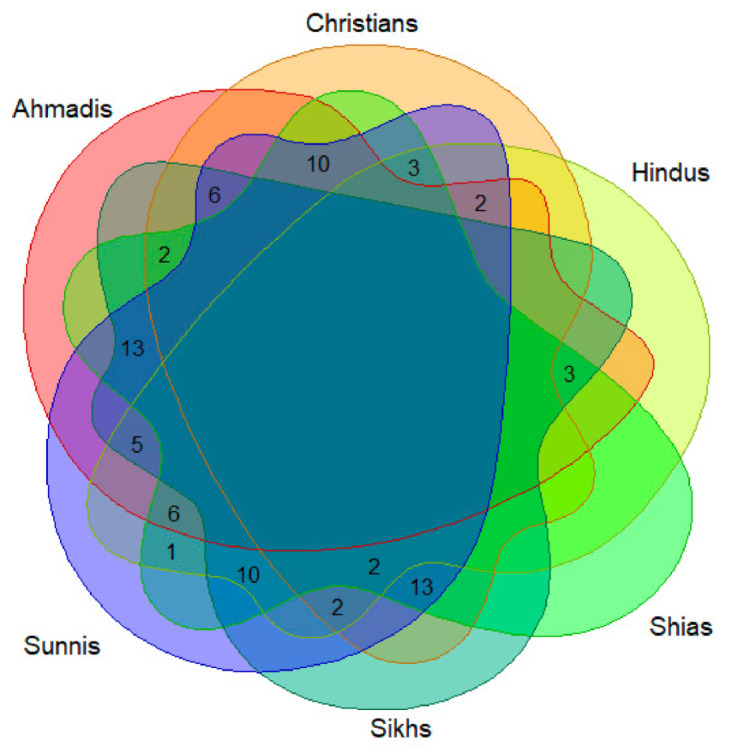
Venn diagram showing the overlaps of the recorded wild food plants among the six considered groups.

**Figure 7 foods-10-00594-f007:**
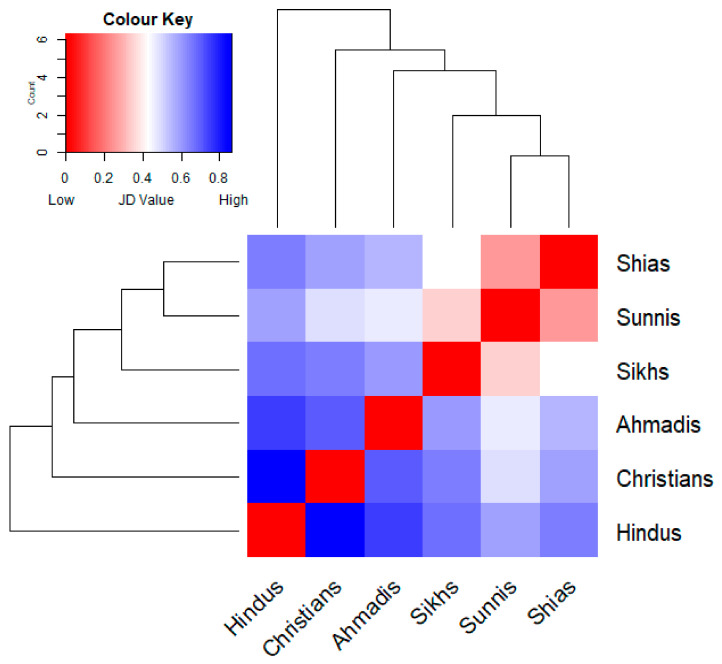
Hierarchical clustering tree coupled with heat map depicting Jaccard Dissimilarity Indices calculated by comparing the wild food plants quoted by the six considered groups.

**Figure 8 foods-10-00594-f008:**
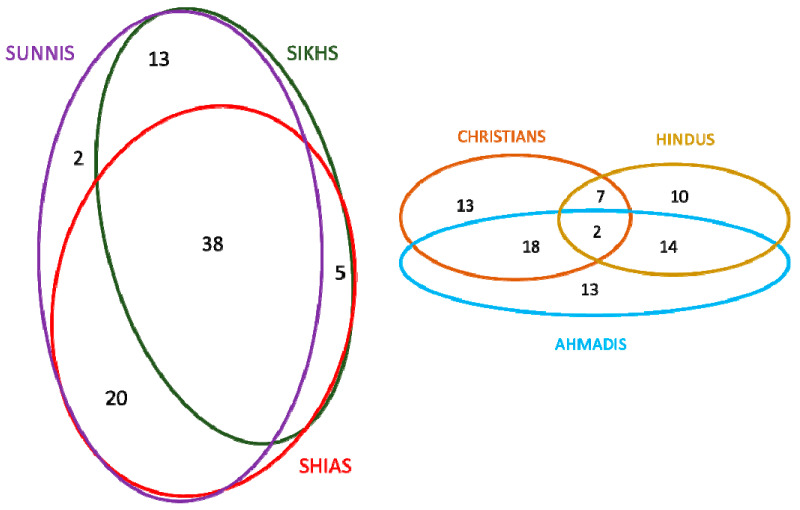
Intuitive best fit Venn diagrams comparing the recorded wild food plants among the six religious groups divided into two clusters.

**Figure 9 foods-10-00594-f009:**
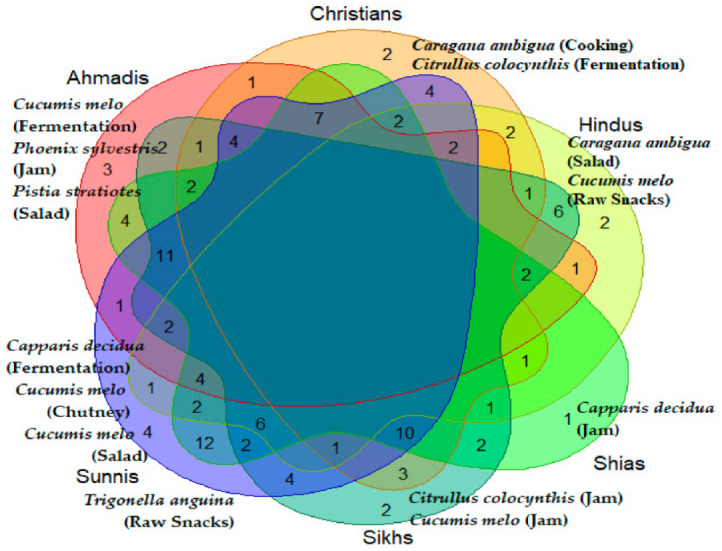
Venn diagram showing the number of overlaps of the recorded wild food plant uses among the studied religious groups; the diagram shows also the food uses uniquely recorded within each group.

**Figure 10 foods-10-00594-f010:**
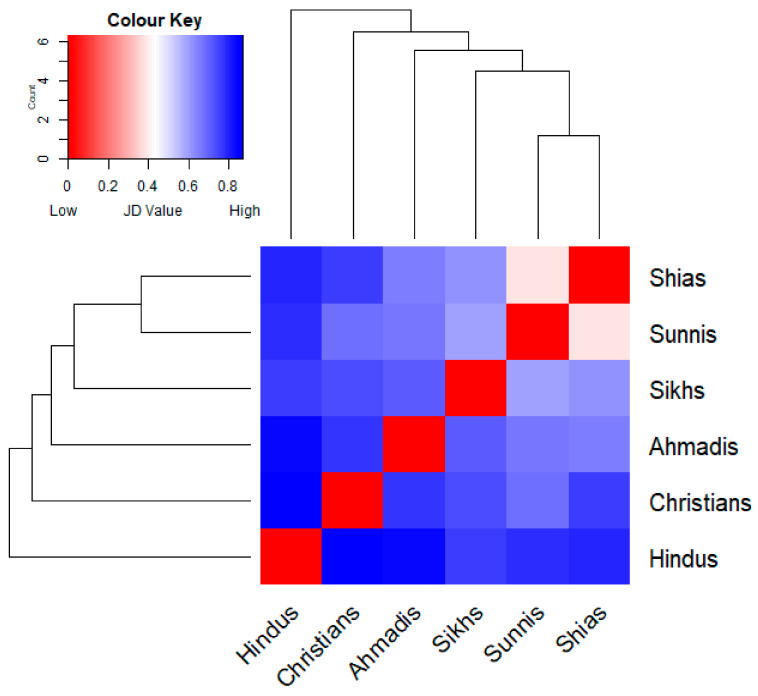
Hierarchical clustering tree coupled with heat map depicting Jaccard Dissimilarity Indices calculated by comparing the actual food utilizations of the recorded wild food plants among the six considered groups.

**Figure 11 foods-10-00594-f011:**
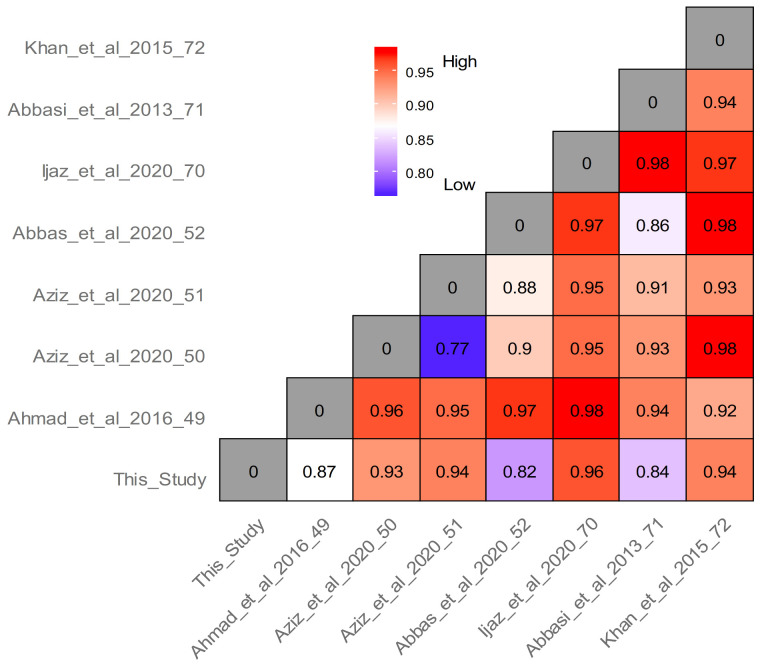
Pairwise Jaccard Dissimilarity Indices calculated by comparing the current study with other wild food ethnobotanical field works previously conducted in Pakistan.

**Table 1 foods-10-00594-t001:** Characteristics of the study participants.

Religious Group	Sunnis	Shias	Hindus	Sikhs	Christians	Ahmadis
Brief historical sketch	Islam arrived in the 8th century, the majority converted to Sunni Islam during in the 11th–16th centuries; minor fractions migrated from Middle Eastern and African countries	The majority converted to Shia Islam during the 16th–18th centuries; minor fractions migrated from the Middle East	Autochthonous	Converted around the 15th century	Emerged with British colonialism in the 18th and 19th centuries	Converted in the 19th century
Approx. number of inhabitants in Jhelum District, Pakistan (2020)	1.01 million	0.21 million	2000	5000	7000	6000
Study villages	Dhoke Padhal, Dharyala, Chakoha, Mohal, Natain, Zinda Shah Madar, Surghdan,	Chak Jamal, Kundal, Pindori, Nathot	Chak Akka, Nathwala, Nakodar, Pari Darweza	Dhaniala, Nougran, Adranah	Nagial, Kot Umar, Dharyala Jalap	Naka Kalan, Rajipur, Kharala, Wara Phophra, Langer Pur
Spoken languages	Pothwari, Kashmiri, Pashto	Saraiki, Pahari, Pothwari	Hindku, Hindi, Sindhi	Punjabi, Gojri	English, Urdu	Urdu
Inter-marriages	Rarely exogamic with Shia	Rarely exogamic with Sunni	Endogamic	Strictly endogamic	Endogamic	Strictly endogamic
Main occupations	Forestry and farming	Forestry and farming	Farming and urban occupations	Pastoralism and urban occupations	Horticulturalism and urban occupations	Horticulturalism
Estimated average socio-economic status of the study participants	Middle	Middle	Low	Low	Low	Middle low
Number of study participants	20	20	20	20	20	20
Percent of female participants	30%	25%	25%	45%	45%	30%
Overall mean age of the study participants	47	53	64	66	59	69

**Table 2 foods-10-00594-t002:** Recorded wild food plants and their local uses.

Plant Species, Family, and Voucher Specimen Number	Local Names	Parts Used	Gathering Area and Season	Local Culinary Uses and Quotation Frequency	Frequency of Consumption
*Acacia modesta* Wall.; Leguminosae; 827/MM//2020	Pholai^UR, PN, PT, HN^ Pali^PH, GJ^ Jangli Kiker^SR^ Palosa^PS^ Angrezi Babur^SN^ Kiker Kul^KM^	Gum, Leaves	DL, FO, HS, RS, SP, WP; March–April	Fermentation^HN**, SI*^ Jam^SH*, SN^	Very common^SH^ Common^HN^ Rare^SN^ Very rare^SI^
*Acacia nilotica* (L.) Delile; Leguminosae; 783/MM//2020	Kikar^UR, PN, PT, HN, HI^ Kikr^SR, PH, GJ, KM^ Kikhar^PS^ Sindhi Babur^SN^	Gum, Pods	DL, FO, HS, RS, SP, WP; March–April	Fermentation^SH**, QA*^ Jam^SI**, SN^	Very common^SH^ Common^QA^ Rare^SN^ Very rare^SI^
*Aerva javanica* (Burm. f.) Juss. ex Schult.; Amaranthaceae; 544/MM//2020	Boen^UR^ Thoo^PN^ Boi^PT, PH, GJ, KM^ Niki Boien^SR^ Shorakai^PS^ Sparokai^PS^ Booh^SN^ Safed Bui^HI^	Flowers, Leaves, Seeds	DL, FO, GR, HS, SP, SL, WP; February–April	Cooking^CR**, QA*, SH, SN**^	Very common^SH^ Common^SN^ Rare^CR^ Very rare^QA^
*Agave americana* L.; Asparagaceae; 675/MM//2020	Jangli Kwar Gandal^UR^ Laphra^PN, PH, GJ^ Kanwar Phara^SR^ Desi Kwar Gandal^PT^ Kamal Gand^KM^ Keuro^SN^ Zargira^PS^ Kamal Cactus^HN^ Bin Katora^HI^	Leaves	AL, DL, FO, GL, GR, RS, SP, WP; August–September	Cooking^SN, SH, HN*, QA **^	Very common^SH^ Common^SN^ Rare^QA^ Very rare^HN^
*Allium carolinianum* DC.; Amaryllidaceae; 409/MM//2020	Jangli Pyaz^UR, KM, HN^ Jangli Ganda^PN, PT, GJ^ Jangli Wasal^SR^ Khokhai^PS^	Bulbs	DL, FO, GL, SP, SH; August–September	Cooking^CR*, SN*^ Salad^QA*, SH*^	Very common^SN^ Common^CR^ Rare^SH^ Very rare^QA^
*Amaranthus spinosus* L.; Amaranthaceae; 787/MM//2020	Cholai^UR^ Konjel^PN^ Surkh Gunahr^PH, PT^ Batto^SR^ Ghinyar^GJ, KM^ Chalvery^HN^ Sarmay^PS^ Kalga^SN^ Gulee^KM^ Ganhar^HN^ Kanta Chaulai^HI^	Leaves	AL, GL, HS, RS, SL, SH, WP; August–September	Cooking^CR*, SH***, SI**, SN^	Very common^SN^ Common^SI^ Rare^CR^ Very rare^SH^
*Amaranthus viridis* L.; Amaranthaceae; 878/MM//2020	Jangli Choolai^UR^ Tandla^PT, GJ, PH^ Tandula^SR^ Tanduli^PN^ Lutur^SN^ Saag^PS^ Ranzaka^PS^ Ganyar^HN^ Ganar^KM^	Leaves	AL, GL, HS, RS, SL, SH, WP; August–September	Cooking^QA*, SH, SI**, SN^	Very common^SN^ Common^SI^ Rare^SH^ Very rare^QA^
*Boerhavia repens* L.; Nyctaginaceae; 816/MM//2020	Looni Booti^UR^ Lornki^PN, PT^ Lorank^SR^ Bakhro^SN^	Leaves	AL, FO, GL, GR, HS, RS, SH, WP; August–September	Cooking^SN**, SH, SI**, CR**^	Very common^SN^ Common^SH^ Rare^CR^ Very rare^SI^
*Cannabis sativa* L.; Cannabaceae; 669/MM//2020	Bhang^UR, SN, SR, GJ, PH, KM^ Pang^PN, PT, HN^ Kamm^PS^	Leaves, Seeds	AL, GL, GR, RS, SH, WP, WL; March–April	Herbal drink^SN, SH*, SI**, QA^	Very common^SN^ Common^SH^ Rare^QA^ Very rare^SI^
*Capparis decidua* (Forssk.) Edgew.; Capparaceae; 532/MM//2020	Karir^UR^ Pichu^UR^ Karinha^PN, GJ^ Kari^SR^ Kareenh^PT, PH, KM^ Kareer^HN^ Dela^GJ, KM^ Kreeta^PT^ Kareenh^SR^ Kira^PS^ Jaba^PS^ Kirar^SN^ KairHI	Fruits	DL, FO, GR, HS, SP; August-September	Fermentation^SN**^ Jam^SH*^ Raw snacks^SI**, CR*^	Very common^SH^ Common^SN^ Rare^CR^ Very rare ^SI^
*Caragana ambigua* Stocks.; Leguminosae; 409/MM//2020	Jangli Phali^UR, PN, PT, GJ^ Baiphli^SR^ Zaray^PS^	Flowers, Pods	RS, SL, SH, WP; June–July	Cooking^CR*^ Raw snacks^SN*, SI^ Salad^HN**^	Very common^SN^ Common^SI^ Rare^CR^ Very rare^HN^
*Chenopodium album* L.; Amaranthaceae; 748/MM//2020	Jangli Bathoo^UR^ Desi Bathoo^PN, PT, PH, GJ^ Desi Battoon^SR^ Surma^PS^ Sormi^PS^ Spin Soba^PS^ Buthia^PS^ Udharam^HN^ Chil^SN^ Bathwa^KM^ Goyalo^HI^	Branches, Leaves	AL, FO, GL, GR, HS, RS, SL, SH, WP; March–April, August–September	Cooking^SN**, SI***, CR*, SH**^	Very common^SN^ Common^SI^ Rare^CR^ Very rare^SH^
*Chenopodium murale* L.; Amaranthaceae; 805/MM//2020	Karnd^UR, PH, GJ, KM^ Karwa Bathoo^PN^ Bathoo^PT^ Kora Batoon^SR^ Thor Surma^PS^ Lulur^SN^ Kurund^HN^ Goyalo^HI^	Branches, Leaves	FO, GL, GR, HS, RS, SL; March–April	Cooking^SN, SH**, CR*, QA^	Very common^SI^ Common^SN^ Rare^SH^ Very rare^QA^
*Chenopodium vulvaria* L.; Amaranthaceae; 611/MM//2020	Sufaid Bathoo^UR, KM^ Jangli Batoon^PN, PT, PH^ Chitta Batoon^SR^ Lulur^SN^ Kurund^HN^ Goyalo^HI^	Branches, Leaves	AL, DL, RS; March–April, August–September	Cooking^SN*, SH, SI*, QA*^	Very common^SI^ Common^SN^ Rare^SH^ Very rare^QA^
*Cirsium arvense* (L.) Scop.; Asteracae; 761/MM//2020	Leeh^UR^ Lehi^PN, PT^ Leh^PH^ Liah^GJ^ Wanvahri^SR^ Da Khwarak Azghai^PS^ Kandiara^SN^ Kund^KM^	Stems	DL, FO, GL, GR, HS, SP, SH, WP; March–April	Raw snacks^SN*, SH***, HN*, QA*^	Very common^SH^ Common^QA^ Rare^HN^ Very rare^SN^
*Citrullus colocynthis* (L.) Schrad.; Cucurbitaceae; 638/MM//2020	Tumma^UR^ Kaud Tumbha^PN, GJ, PH^ Kor Tumma^PT, SR, KM^ Pirpandyan^PS^ Marghone^PS^ Tarha Marha^PS^ Andrain^PS^ Hanzal^PS^ Trooh^SN^ Indrayan^HI^	Fruits	AL, DL, FO, GL, GR, HS, RS, SP, SL, SH, WP; May–June	Fermentation^CR***^ Jam^SI**^ Spice^SN*, SH***^	Very common^SI^ Common^SN^ Rare^CR^ Very rare^SH^
*Commelina benghalensis* L.; Commelinaceae; 795/MM//2020	Kani^PN, SR^ Jawarzaal^PS^ Chura^KM^	Leaves	FO, GL, HS, RS, SL, SH, WP, WL; March–April	Cooking^SN, SH*, SI***, QA**^	Very common^SH^ Common^SN^ Rare^SI^ Very rare^QA^
*Convolvulus arvensis* L.; Convolvulaceae; 728/MM//2020	Lehi^UR^ Lehli^PN, GJ^ Hiran Kahri^PT, PH^ Vanvaihre^SR^ Parvaty^PS^ Naaro^SN^ Speaker Booti^HN^ Hirapadi^KM^	Leaves	AL, FO, GL, GR, RS, SH, WL; March–April	Cooking^SH, SI**, CR**, QA^	Very common^SH^ Common^QA^ Rare^CR^ Very rare^SI^
*Corchorus depressus* (L.) Stocks; Malvaceae; 591/MM//2020	Bahu Phali^UR^ Baephli^SR^ Munderi^SN^	Whole plant	DL, GR, HS, MS, SP, SL; March–April	Herbal drink^SN, SI***, CR**, QA*^	Very common^SI^ Common^QA^ Rare^SN^ Very rare^CR^
*Corchorus tridens* L.; Malvaceae; 417/MM//2020	Phali^UR, PN, PT, GJ, KM^ Dadi^SR^	Pods	GL, GR, HS, MS; March–May	Herbal drink^SN, SH*, SI*, QA*^	Very common^SH^ Common^QA^ Rare^SN^ Very rare^SI^
*Cucumis melo* L.; Cucurbitaceae; 527/MM//2020	Chibar^UR, PN^ Chibbarh^SR^ Chibhar^PH, PT, GJ, KM^ Mitero^SN^	Fruits	AL, GL; June–July	Chutney^SN***^ Fermentation^QA**^ Jam^SI*^ Raw snacks^HN***^ Salad^SN***^	Very common^SI^ Common^HN^ Rare^QA^ Very rare^SN^
*Digera muricata* (L.) Mart.; Amaranthaceae; 694/MM//2020	Tandla^UR^ Tandoli^PT, GJ, PH^ Leswa^KM^ Tandala^PN^ Mareeri Saag^SR^ Athi^HN^ Tartara^PS^ Nazam Hoora^PS^ Lulur^SN^ Chanchali^HI^ Lahsuva^HI^	Branches, Leaves	FO, GL, GR, HS, RS, SH, WP, WL; August–September	Cooking^SI*, CR**, SN*, QA^	Very common^SI^ Common^SN^ Rare^CR^ Very rare^QA^
*Dysphania ambrosioides* (L.) Mosyakin & Clemants; Amaranthaceae; 856/MM//2020	Desi Bathoo^UR^ Bathoo^PN, PT^ Jangli Battoon^SR^ Babre Nagdi^PS^ Bathu^GJ, PH^ Bathwa^HN, KM^	Branches, Leaves	AL, DL, FO, GL, GR, HS, RS, SL, SH, WP; August–September	Cooking^SH**, CR**, SN*, QA**^	Very common^SH^ Common^SN^ Rare^CR^ Very rare^QA^
*Fagonia indica* Burm. f.; Zygophyllaceae; 842/MM//2020	Jamahon^UR, PN, PT^ Damanh^PN, KM, GJ^ Jawanh Booti^SR^ Dramaho^SN^	Whole plant	DL, FO, GR, HS, RS, SP, SL; July–August	Herbal drink^SN*, SH, SI**, QA^	Very common^SH^ Common^SI^ Rare^QA^ Very rare^SN^
*Galium aparine* L.; Rubiaceae; 589/MM//2020	Wanwair^PN, PT, GJ^ Wanwair Booti^SR, PH^ Cochna^PS^ Lahndra^KM^	Leaves	AL, FO, GL, GR, HS, RS, SL, SH, WP; June–July	Herbal Drink^SN**, SI, CR, QA^	Very common^SI^ Common^QA^ Rare^SN^ Very rare^CR^
*Gisekia pharnaceoides* L.; Gisekiaceae; 644/MM//2020	Balu Ka Sag^UR^ Jangli Sag^PN, PT, SR^	Leaves	AL, FO, GL, GR, RS, SP, SH, WP; July–August	Cooking^SN*, SH**, SI**, CR**^	Very common^CR^ Common^SH^ Rare^SN^ Very rare^SI^
*Indigofera hochstetteri* Baker.; Leguminosae; 499/MM//2020	Kano^UR^ Raari^PN, PT, GJ, PH^ Mareeri^SR^ Zind^KM^ Jhill^SN^	Flowers, Seeds	GL, GR, HS, MS; August–October	Jam^SH, SI*, CR*, QA^	Very common^SI^ Common^QA^ Rare^SH^ Very rare^CR^
*Lathyrus aphaca* L.; Leguminosae; 844/MM//2020	Jangli Matter^UR, PN, PT^ Jangli Mattri^SR^ Marghayo Hpay^PS^ Kukarmany^PS^ Jangli Phali^KM^	Pods	AL, FO, GL, RS, SH, WP, WL; September–October	Fermentation^HN**, SI***^ Raw snacks^QA**, SH***^	Very common^SH^ Common^QA^ Rare^HN^ Very rare^SI^
*Lathyrus sativus* L.; Leguminosae; 572/MM//2020	Jangli Matter^UR, PN, PT^ Jangli Mattri^SR^ Marghayo Hpay^PS^ Kukarmany^PS^ Jangli Matar^SN^	Pods	AL, FO, GL, HS, RS, WP, WL; March–April	Cooking^CR**, SN**^ Raw snacks^SH*, SI**^	Very common^SN^ Common^SH^ Rare^CR^ Very rare^SI^
*Launaea procumbens* (Roxb.) Ramayya & Rajagopal; Asteracae; 821/MM//2020	Dodak^UR, PN^ Bhathala^PT^ Hund^PH, GJ, KM^ Dodhk^SR^ Sondrashi^PS^ Alakoo^PS^ Bhattar^SN^	Leaves	AL, FO, GL, GR, RS, SH, WP, WL; March–April	Raw snacks^SN***, SI*, CR**, QA^	Very common^SI^ Common^SN^ Rare^QA^ Very rare^CR^
*Lepidium apetalum* Willd.; Brassicaceae; 505/MM//2020	Jangli Khoob Kalan^UR^ Bashky^PS, PH, PT^ Desi Halyun^SR^ Burchan^HN^ Hanon^PS^ Harf^PS^ Haleem^PS^	Leaves	FO, GL, HS, MS, SH, WP, WL; July–August	Cooking^SN, CR**, HN*, QA**^	Very common^SN^ Common^CR^ Rare^QA^ Very rare^HN^
*Lepidium draba* L.; Brassicaceae; 459/MM//2020	Senna^UR^ Suchi Senna^PN, PH, GJ, PT^ Koori Sana^SR^ Ghora Wal^SN^ Dadhwal^SN^	Leaves, Seeds	DL, FO, GR; April–July	Raw snacks^QA*, SI*^ Salad^HN*, SN*^	Very common^SN^ Common^QA^ Rare^HN^ Very rare^SI^
*Malva neglecta* Wallr.; Malvaceae; 665/MM//2020	Sitara Sunchal^PN, SR^ Tikalay^PS^ Jungali Soxal^KM^ Sonchal^UR, HN, PT, PH^ Khubasi^HI^	Leaves	AL, DL, FO, GL, GR, RS, SH, WP, WL; March–April	Cooking^SN**, SH***, SI*, QA*^	Very common^SI^ Common^QA^ Rare^SN^ Very rare^SH^
*Malva parviflora* L.; Malvaceae; 510/MM//2020	Jangli Sonchal^UR, PN, PT, HN^ Jungali Soxal^KM^ Jangli Khubasi^HI^	Fruits	AL, DL, FO, GL, GR, RS, SH, WP, WL; March–April	Cooking^SH, QA*^ Herbal tea^CR*, SN**^	Very common^SH^ Common^QA^ Rare^CR^ Very rare^SN^
*Malva sylvestris* L.; Malvaceae; 564/MM//2020	Jamni Phool^UR^ Methrai^PN, PS, SR^ Khawazamary^PS^ Samchal^PT, PH, KM^ Khabazi^HN^	Leaves	RS, SH; April–May	Cooking^SH**, SI***, QA, SN**^	Very common^SN^ Common^SI^ Rare^SH^ Very rare^QA^
*Mentha arvensis* L.; Lamiaceae; 693/MM//2020	Podina^UR, PN, PT, PH^ Podna^SR^ Shinshobai^PS^ Podina^GJ, HN^	Leaves	AL, GL; March–April, August–September	Chutney^SH***, SN***, SI**^ Cooking^SH*, SN**, SI**, CR*^ Herbal tea^HN**, SI**^ Spice^CR**, SH*, SN*^	Very common^HN^ Common^CR, SI^ Rare^SI^ Very rare^SH^
*Mentha longifolia* (L.) L.; Lamiaceae; 698/MM//2020	Jangli Podina^UR, PN, KM^ Chita Podna^SR, HN, PT, PH^ Vaylanai^PS^ Shinshobai^PS^ Bareena^SN^	Leaves	FO, GL, HS, SL, SH, WL; May–June, August–September	Chutney^SN**, SH**^ Cooking^SN*, SH*^ Herbal tea^SI***, CR**^ Spice^CR*, SI**, QA**^	Very common^SI^ Common^CR^ Rare^QA^ Very rare^SN^
*Mentha pulegium* L.; Lamiaceae; 659/MM//2020	Jamni Podina^UR, PN, PT, PH^ Desi Podna^SR^ Pudina^KM^	Leaves	AL, FO, GL, HS, RS, SL, SH, WL; March–April, August–September	Chutney^HN*, SI**^ Herbal tea^QA**, SN**^	Very common^HN^ Common^SN^ Rare^SI^ Very rare^QA^
*Mentha royleana* Wall. ex Benth.; Lamiaceae; 631/MM//2020	Sofaid Podina^UR, PN, PT, PH^ Chitta Podna^SR, HN^ Jangli Podina^KM^	Leaves	AL, FO, GL, HS, RS, SL, SH, WL; March–April	Chutney^SH**, SN*^ Cooking^SH*, SN*^ Herbal tea^CR***, HN*^	Very common^SN^ Common^SH^ Rare^HN^ Very rare^CR^
*Olea europaea* subsp. *cuspidata* (Wall. & G. Don) Cif.; Oleaceae; 746/MM//2020	Kahou^UR, PN, PT^ Kao^GJ, KM, PH, SR^ Shwawan^PS^ Khuna^PS^ Kaow^HN^ Kaho^HN^	Fruits	AL; August–September	Raw snacks^QA**, SH**, SI*, HN*^	Very common^SH^ Common^SI^ Rare^QA^ Very rare^HN^
*Opuntia dillenii* (Ker Gawl.) Haw.; Cactaceae; 699/MM//2020	Khashi^UR^ Thor^PH, GJ, KM^ Peeli Saroon^PN^ Peela Saroon^PT^ Peela Rayea^SR^ Woraki^PS^ Shersham^PS^ Hoob Sublan^PS^ Hakseer^PS^	Leaves	AL, GL, HS, RS, SH, WP; March–April	Cooking^SN**, SH*, HN*, QA^	Very common^HN^ Common^SH^ Rare^QA^ Very rare ^SN^
*Oxalis corniculata* L.; Oxalidaceae; 732/MM//2020	Peeli Booti^UR^ Choti lonak^PN^ Lonak^SR^ Therwashka^PS^ Bibi Shaftala^PS^ Tarookay^PS^ Khati Buti^HN^ Khati^KM^	Leaves	AL, FO, GL, GR, HS, RS, SH, WP, WL; February–March	Chutney^CR**, QA*^ Cooking^SI**, SN**^	Very common^CR^ Common^SI^ Rare^QA^ Very rare^SN^
*Phoenix sylvestris* (L.) Roxb.; Arecaceae; 501/MM//2020	Jangli Khajoor^UR, PN, PT, HN^ Desi Khajoor^GJ, PH, KM^ Pind^SR^ Chotti Khagoor^PS^ Khaji^SN^ Khajur^HI^	Fruits	AL, DL, GL, RS; June–July	Jam^QA*^ Raw snacks^SN*, HN, SH^	Very common^SH^ Common^SN^ Rare^HN^ Very rare^QA^
*Physalis divaricata* D. Don; Solanaceae; 569/MM//2020	Jungli Berry^UR^ Jangli Tamator^PN, SR^ Hundusi^GJ, PT, PH, KM^ Band Malkhovj^PS^ Delhuu^SN^	Fruits	FO, GL, HS, SL, SH, WP; August–September	Raw Snacks^SN*, SI*, HN**, QA**^	Very common^SI^ Common^QA^ Rare^HN^ Very rare^SN^
*Pistia stratiotes* L.; Araceae; 515/MM//2020	Jall Khumbi^UR, HI, KM^ Jall Shamkala^GJ, PT, PH^ Nargis^PN, PT^ Jaru^SN^	Leaves	WP; March–April	Cooking^SN*, SH, SI***^ Salad^QA*^	Very common^SN^ Common^SI^ Rare^SH^ Very rare^QA^
*Polygonum plebeium* R.Br.; Polygonaceae; 531/MM//2020	Gorakh Pan^UR^ Droonk^PN^ Bandoki^PS^ Gull Srah^PS^ Khowar^SN^ Chimati Saag^HI^	Stems	AL, FO, GL, GR, HS, MS, RS, SL, SH, WP, WL; March–April	Cooking^SN**, SI**, CR*, HN^	Very common^SI^ Common^CR^ Rare^HN^ Very rare^SN^
*Portulaca oleracea* L.; Portulacaceae; 865/MM//2020	Kulfa Lonak^UR, PN^ Lorniki Booti^PT, GJ, KM^ Lorni Booti^SR^ Varhori^PS^ Loonk^SN^ Khurfa^HN^	Leaves, Stems	FO, GL, HS, RS, SH, WP, WL; August–September	Cooking^SN**, SH*, HN*, QA**^	Very common^SH^ Common^HN^ Rare^SN^ Very rare^QA^
*Portulaca quadrifida* L.; Portulacaceae; 753/MM//2020	Lornak Booti^UR, PN^ Loranki^PT, GJ, PH^ Lonak^SR^ Wakhorai^PS^ Pakharai^PS^ Loonk^SN^ Lunak^KM^ Kolfa^HN^	Leaves, Stems	FO, GL, GR, HS, MS, RS, SL, SH, WP; August–September	Cooking^SH**, SN*, CR*, QA^	Very common^CR^ Common^SN^ Rare^SH^ Very rare^QA^
*Prosopis cineraria* (L.) Druce; Leguminosae; 745/MM//2020	Jand^UR, PN, PT, PH, KM^ Jandi^SR^ Kandi^SN^ Jangli Matar^KM^ Jhand^HI^ Khejri^HI^	Gum, Pods	DL, FO, GR, HS, RS, SP, SL, WP; August–September	Fermentation^SN*, CR**^ Jam^QA**, SH*^	Very common^QA^ Common^SH^ Rare^CR^ Very rare^SN^
*Prosopis juliflora* (Sw.) DC.; Leguminosae; 547/MM//2020	Kikar^UR^ Phari Kikar^PN, PT, GJ, KM, SR^ Sindhi Kikar^PH^ Kikar^PS^ Angrezi Babur^SN^ Velayti Kikar^HN^ Jungli Kikar^HI^	Gum, Pods	AL, FO, GR, HS, RS, WP; August–September	Fermentation^SI, HN*, CR^ Jam^SH**, SN*^	Very common^SH^ Common^SN^ Rare^HN^ Very rare^SI, CR^
*Rhynchosia minima* (L.) DC.; Leguminosae; 855/MM//2020	Jangli Lobia^UR, PN, PH, KM^ Jangli Arwan^PT^ Herdal^SR^	Pods	AL, FO, HS, RS, WP, WL; March–April	Cooking^SN, SH*, SI**, CR^	Very common^CR^ Common^SI^ Rare^SN^ Very rare^SH^
*Rumex dentatus* L.; Polygonaceae; 812/MM//2020	Khatkal^PN, PT, PH, KM^ Jangli Palak^UR, GJ, SR^ Sarkari Palak^PS^ Zamda^PS^ Jangli Palak^SN^ Hullah^HN^ Ola^HN^	Leaves	AL, FO, GL, HS, RS, SL, SH, WP, WL; March–April	Cooking^SN**, SH**, SI**, HN^	Very common^SI^ Common^HN^ Rare^SH^ Very rare^SN^
*Salvadora oleoides* Decne.; Salvadoraceae; 690/MM//2020	Jall^UR^ Van^GJ, KM^ Jhal^PT, PH^ Pilu^PN,SR^ Khabbar^PS^ Khabar^SN^ Kallijari^HN^	Fruits	DL, FO, GR, HS, SP, WP; August–September	Chutney^HN*, CR*^ Fermentation^SH*, SN**^ Jam^SH*, HN*, CR*^	Very common^SH^ Common^SN^ Rare^CR^ Very rare^HN^
*Salvadora persica* L.; Salvadoraceae; 747/MM//2020	Pelo^UR, SR, GJ^ Khabar^SN^ Pilu^PN, PT, PH, KM^ Diyar^SN^ Kallijari^HN^ Jaal^HI^	Fruits	DL, GR, RS, SP, SL, WP; August–September	Fermentation^SH*, SN*^ Jam^SI*, HN**^	Very common^SH^ Common^SN^ Rare^SI^ Very rare^HN^
Salvia moorcroftiana Wall. ex Benth.; Lamiaceae; 530/MM//2020	Tokham Belaga^UR^ Lapra^PN^ Belangoo^SR^ Dersai^PS^ Sidrai^PS^ Jungle Tamookh^KM^ Shwanko^SN^ Kallijari^HN^ Khesari Daal^HI^	Stems	FO, HS, SL, SH, WP; May–June	Raw snacks^SN***, SH, QA **, CR*^	Very common^SH^ Common^QA^ Rare^CR^ Very rare^SN^
*Salvia nubicola* Wall. ex Sweet; Lamiaceae; 841/MM//2020	Hernar^PN^ Darshool^PS^ Kallijari^HN^ Khesari Daal^HI^	Leaves	FO, HS, RS, SH, WP, WL; August–September	Cooking^SH**, SI*, HN***, QA^	Very common^SI^ Common^SH^ Rare^HN^ Very rare^QA^
*Senna italica* Mill.; Leguminosae; 479/MM//2020	Ghora Wal^SN^	Seeds	GL, GR, HS, MS, RS; April–June	Raw snacks^SN*, SH*, SI, QA**^	Very common^SN^ Common^QA^ Rare^SH^ Very rare^SI^
*Senna occidentalis* (L.) Link; Leguminosae; 576/MM//2020	Lobia^UR, PN, GJ, KM^ Desi Arwan^SR, PT, PH^ Ghora Wal^SN^	Pods	AL, FO, HS, RS, WP, WL; March–April	Cooking^SN**, SH, CR*, QA^	Very common^CR^ Common^SN^ Rare^QA^ Very rare^SH^
*Sisymbrium irio* L.; Brassicaceae; 750/MM//2020	Khud-e-Kalan^KM^ Khashi^UR, PN, PT, PH^ Peeli Booti^SR^ Woraki^PS^ Shersham^PS^ Hoob Sublan^PS^ Hakseer^PS^ Khubkalan^HN^ Khakasi^HN^	Leaves	AL, GL, HS, RS, SH, WP; March–April	Cooking^SN, SH**, SI***, HN**^	Very common^SI^ Common^SN^ Rare^HN^ Very rare^SH^
*Solanum americanum* Mill.; Solanaceae; 636/MM//2020	Makao^UR^ Kainch Mainch^PN^ Katch Match^PT^ Mohkri^PH, GJ, KM^ Karveloon^SR^ Kach machao^PS^ Malkhovj^PS^ Malgabai^PS^	Fruits	AL, FO, GL, GR, HS, RS, SH, WP; June–July	Chutney^SI*, SH^ Herbal drink^SN**, SI^ Raw snacks^HN*, SH, SI*, SN**^	Very common^SH^ Common^SI^ Rare^HN^ Very rare^SN^
*Solanum incanum* L.; Solanaceae; 727/MM//2020	Jangli Khashi^UR^ Jangli Baingan^PN, PT^ Mahokari^PS^ Kori Wal^SR^ Mahora^SN^	Fruits	FO, GL, HS, SH, WP; June–July	Chutney^SH, SN*^ Raw snacks^QA**, HN**^	Very common^SH^ Common^QA^ Rare^HN^ Very rare^SN^
*Solanum surattense* Burm. f.; Solanaceae; 758/MM//2020	Neeli Khurd Katai^UR^ Choti Kandiari^PN^ Mahori^PT, GJ, PH, KM^ Kandiari Walh^SR^ Markondaye^PS^ Speenazghai^PS^ Kanderi^SN^ Mohkree^HN^	Fruits	DL, FO, GR, HS, MS, RS, SP, SL; October–November	Raw Snacks^SN, CR*, HN*, QA*^	Very common^HN^ Common^QA^ Rare^CR^ Very rare^SN^
*Solanum villosum* Mill.; Solanaceae; 415/MM//2020	Mako^UR, SN^ Kaach Mach^PN, PT, GJ, KM^ Karveloon^SR^	Fruits	GL, GR, HS, MS; March–May	Chutney^SI*, HN*^ Raw snacks^SH**, SN**^	Very common^SI^ Common^SH^ Rare^SN^ Very rare^HN^
*Sonchus asper (*L.) Hill; Asteracae; 666/MM//2020	Bhattal^UR^ Malai Booti^PN^ Dodhi^PT, PH, GJ^ Dodhak^SR^ Soon Latti^PS^ Kasni^SN^ Dodal^KM^	Leaves	DL, FO, GL, HS, RS, SL, SH; March–April	Cooking^SH**, HN***, SN^	Very common^SH^ Common^SN^ Very rare^HN^
*Sonchus oleraceus* (L.) L.; Asteracae; 713/MM//2020	Bhattal^UR^ Malai Booti^PN^ Dodhak^PT^ Peeli Dodhak^SR^ Tarizha^PS^ Soon Dodak^PS^ Kasni^SN^	Leaves	AL, FO, GR, RS, SH; March–April	Cooking^SN*, SI**, HN***, QA**^	Very common^SI^ Common^QA^ Rare^HN^ Very rare^SN^
*Stellaria media* (L.) Vill.; Caryophyllaceae; 796/MM//2020	Kangni Booti^UR^ Phoolan Cheeri^PN^ Cheeri Pta^PT^ Stalli^PH, GJ, KM^ Chitti Booti^SR^ Vilaghori^PS^ Badsha Saba^PS^ Bin Batorhi^PS^ Buch-Bucha^HI^	Leaves	AL, FO, GL, HS, MS, RS, SL, SH, WP; March–April	Cooking^SN***, SH**^ Herbal tea^CR**, SI^	Very common^SH^ Common^SN^ Rare^CR^ Very rare^SI^
*Tephrosia purpurea* (L.) Pers.; Leguminosae; 429/MM//2020	Bansa-Bansu^PN, PT, PH, GJ, KM^ Sarphooka^PS^ Haldri^SR^ Maheero^SN^ Ban Nil^HI^	Pods	GL, GR, HS; March–May	Cooking^SN**, SH*, SI, QA*^	Very common^SI^ Common^SH^ Rare^QA^ Very rare^SN^
*Tribulus terrestris* L.; Zygophyllaceae; 539/MM//2020	Pakhra^PS, PT, KM^ Bhakhra^UR, SR^ Bakhro^SN^ Melai^PS^ Ghokru^HN^	Fruits	DL, FO, GL, GR, HS, MS, RS, SP, SL, SH, WP; August–September	Herbal tea^SN*, SH, SI**, HN*^	Very common^SH^ Common^SI^ Rare^HN^ Very rare^SN^
*Trigonella anguina* Delile; Leguminosae; 568/MM//2020	Jangli Meethre^UR, PN, PT,GJ, SR^ Jungle Math^KM^	Leaves, Seeds	AL, DL, FO, GL, GR, HS, RS, SH, WP, WL; March–April	Fermentation^SH**, HN*, SI*^ Raw snacks^SN^	Very common^SH^ Common^HN^ Rare^SN^ Very rare^SI^
*Trigonella corniculata* Sibth. & Sm.; Leguminosae; 615/MM//2020	Meethre^UR, PN, PT,GJ, SR^ Jungle Math^KM^	Leaves, Seeds	AL; March–April	Fermentation^QA*, CR, SN*, SH*^	Very common^CR^ Common^QA^ Rare^SN^ Very rare^SH^
*Veronica anagallis-aquatica* L.; Plantaginaceae; 834/MM//2020	Hazar Dani^UR^ Obo Saba^PS^	Leaves	AL, FO, GL, HS, RS, SH, WP; March–April	Cooking^SN***, SH, SI*, QA*^	Very common^QA^ Common^SI^ Rare^SH^ Very rare^SN^
*Vicia sativa* L.; Leguminosae; 767/MM//2020	Jangli Lobia^UR^ Jangli Rewari^PN, GJ^ Jangli Rawan^SR^ Mutri^KM, PT, PH^ Pervatha^PS^ Chilow^PS^	Pods	AL, FO, GL, HS, SL, SH, WP, WL; March–April	Cooking^SN**, SH*, SI**, CR**^	Very common^SN^ Common^SH^ Rare^CR^ Very rare^SI^
*Withania coagulans* (Stocks) Dunal; Solanaceae; 741/MM//2020	Paneer^UR^ Jangly Chana^PT^ Akri^PN, PH, SR^ Khamzora^PS^ Ashwgandhas^SN^ Asgandh Nagori^SN^	Leaves, Fruits	DL, FO, GL, SP, SH; March–April	Herbal drink^SN, SH, SI*, CR^	Very common^SH^ Common^SI^ Rare^CR^ Very rare^SN^
*Ziziphus jujuba* Mill.; Rhamnaceae; 726/MM//2020	Bairi^UR, PN^ Seo Bair^PT, SR^ Jand Beri^PH, GJ^ Bera^PS^ Moti Ber^PS^ Karkanra^PS^ Ber^SN^ Ber^KM^	Fruits	DL, FO, GL, GR, HS, RS, SP; August–September	Raw snacks^SN**, HN***, SI**, SH**^	Very common^SI^ Common^HN^ Rare^SN^ Very rare^SH^
*Ziziphus nummularia* (Burm. f.) Wight & Arn.; Rhamnaceae; 612/MM//2020	Jangli Bairi^UR, PN, GJ^ Kathy Beer^PT, SR^ Karkanr^PS^ Chotti Ber^PS^ Anane^PS^ Bada Bera^PS^ Jhangugli Ber^SN^ Jahri Ber^HN^	Fruits	FO, GL, HS, RS, SP; April–May	Raw snacks^SN**, HN***, CR**, SH**^	Very common^SH^ Common^HN^ Rare^SN^ Very rare^CR^
*Ziziphus oxyphylla* Edgew.; Rhamnaceae; 409/MM//2020	Surkh Bair^UR, PN^ Saib Bair^SR, PH, PT, GJ^ Heilaneiy^PS^ Phitni^HN^	Fruits	FO, GL, HS, RS, SP; April–May	Raw snacks^SN*, SH*, SI***, CR*^	Very common^SI^ Common^CR^ Rare^SN^ Very rare^SH^
*Ziziphus spina-christi* (L.) Desf.; Rhamnaceae; 413/MM//2020	Jangli Bair^UR^ Jhar Beri^PN, PT, GJ, KM^ Jangali Bair^SR^ Ber^SN^	Fruits	GL, GR, HS, MS, RS; March–June	Raw snacks^SN*, SI*, SH*, CR*^	Very common^CR^ Common^SH^ Rare^SN^ Very rare^SI^
*Coprinus comatus* (O.F. Müll.) Pers.; Agaricaceae; 400/MM//2020	Khumbhi^UR, PN, PT, GJ, SR^ Guchi^PS^ Klikichok^PS^	Arial parts	GL, GR, HS; August–September	Cooking^SN*, SH, SI*, HN**^	Very common^SI^ Common^SH^ Rare^SN^ Very rare^HN^

Gathering areas: AL: arable land, DL: dry land, FO: forest, GL: grassland, GR: graveyard, HS: hilly slopes, MS: mountain summits, RS: roadside, SP: sandy places, SL: scrubland, SH: shady places, WP: paste places, WL: wet land; Local Languages: UR: Urdu, PN: Punjabi, PT: Pothwari, PH, Pahari, GJ, Gojri, HN: Hindko, SR: Saraiki, SN: Sindhi, PS: Pashto, KS: Kashmiri, HI: Hindi; Religious faith: SN: Sunnis, SH: Shias, SI: Sikhs, HN: Hindus, CR: Christians, QA: Ahmadis (Qadiani); Quotation frequency in percent: 1–25% = without asterisk, 26–50% = *, 51–75% = **, 76–100% = ***.

## Data Availability

The data presented in this study are available on request from the corresponding author. The data are not publicly available due to avoid any inter-controversies among the studied religious groups.

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
