# Peer review of "Gathered Wild Food Plants among Diverse Religious Groups in Jhelum District, Punjab, Pakistan"

_foods, 2021, doi:10.3390/foods10030594_

Round 1

Reviewer 1 Report

 Title: Gathered wild food plants among diverse religious groups in Jhelum District, Punjab, Pakistan 
by:  Muhammad Majeed et al.

This manuscript is a well-researched and generally very well written, comprehensive study that leaves relatively little to be criticised. The Introduction with its short and precise sentences does not exactly reflect a 'fluent' and cohesive style, but it is getting the message across that the study is novel and important. However, starting on line 59 and then following through lines 69 there is some repetition (also with information given on line 49). Perhaps the authors could adress that issue in their revision.

What did surprise this reviewer, however, was that no mention of food taboos or temporary deviations from the 'regular' use of plants (wild or otherwise)was made. For Hindus, at least, it has been reported that some plant products will not be eaten on certain days and women, those in a state of pregnancy for example, will abstain from consuming certain plants (see Meyer-Rochow 2009: Journal of Ethnobiology and Ethnomedicine). In a revision this aspect of plant uses by followers of different religions should be mentioned and the authors are advised to explore this issue or to explain why they felt that issue was not worth studying.

Are 'pickles' not a fruit category that is relished in the area by at least some? It surprises me. Are there special events, festivals, religious ceremonies when certain wild plants and dishes made from them are recommended and do the results reported in this study apply to all genders and age groups equally?

The illustrations are fine, but in figures 1, 4, and 5 the individual photos of the composite should be delineated by narrow white borders to make the photos stand out more clearly.

Regarding figures 6 and 9, the colour coding used in the figures should be made obvious in the figure legend rather than in the diagrams themselves. Their legends could have been somewhat more extensive and informative without having to resort to the paper's body of text.

In figure 9 under the label 'Christians' and next to the label 'Hindus' one sees in the diagram the numbers 2 and 2, respectively, but in figure 6 there are no numbers at all seen in the respective places. 

Reviewer 2 Report

Dear authors,

Thank you very much for this thoroughly worked out and detailed article!
I liked the authors' small meta-analysis by comparing plant species lists with previous published articles.
Also, thanks for the balanced sampling approach, something other researchers do often not mind or do not account for during analyses and interpretation!

General
- more emphasis on Table (for instance Table 2) and Figure (for instance Figure 1) captions. The reader should understand at a glance what is displayed WITHOUT cross-checking the text!
- references are inconsistent: (1) decide for capitalized titles of journals or not (check author guidelines of FOODS). (2) capitalisation of geographical and proper names terms such as Pind, Dadan, Khan, Punjab, or Pakistan (for instance REF: 57), (3) scientific names must be italized
- I feel that there should be a more critical reflection on the ethnic composition and the use of species. There are many more factors as those presented, and difference may be just based on the sampling range, the sample size, the region, or how close interviewees have been selected, among other. Just an example, because I know the paper: Aziz, M.A.; Abbasi, A.M.; Ullah, Z.; Pieroni, A. 2020 found the opposite: Shia/Sunni were less related to wild food plants than the local Ismaili/Shia group. What I want say: a bit more distant, and one will do inverse observations. However, you worked out well that species use is rather homogenous than ethnic-/religious-specific. The question however remains what this actually implies for food security and biocultural heritage as you have written in your last sentence of your abstract (line 38)? Is it helpful to use this knowledge in order to draw some measures?

Specific
Line 40: Keywords - no need to mention "wild food plants", "Punjab" and "Pakistan" as they are already present in your title. You loose, let's says, degrees of links to other research by just repeating terms already used in the title. Rather use other words such as "edible plant", "bio-cultural heritage", etc
Line 68: "attached to natural their resources" wording!!
Line 75: one sentence, is not yet one paragraph! Also, the two sub sentences here are actually thematically not the same "socio-economic, cultural, political and economic factors" & "fermentation of local wild food plants" 
Line 119: figure 1 - Please provide a full description (plant community, main species, soils, source rock) to the pictures (a-h) and do not leave the reader with a collage of something! I think also that some urban context should be displayed and not pure nature!
Line 122-123: avoid content-poor sentences as "Some typical landscapes are shown in 122 Figure 1." write instead a description of typical scenarios/landscapes and add afterwards "(Figure 1)" . Same for other tables (for instance Line 134) and figures!
Lie 137: "the National language" = "the national language"
Line 145: I think "following [61]." must be written as "following AUTHOR et al.[61]." check other entries in the text (for instance Line 230: "by [83]." and Line 249)!
Line 192: Figure 3 - Please no 3D display! x- and y-axes have no label (x: "Food plants", y: Percent %)!
Line 216: resolution of pictures should be higher!
Line 232-234: repetition to lines 212-214?
Line 269: "wild food plant gathering" or "wild food plant species"?
Line 276: Figure 7 - is the pure red tone really an attribute or should it be blank/white/black/pattern (as for instance in Figure 11)? Also, please explain the turquois coloured frequency line. I doubt that this is easily understandable by all readers! Would the colour key not better fit if is vertically displayed - particularly concerning the frequency (turquois) line! All points mentioned are the same for Figure 10.
Line 324: "Jaccard's similarity", before you talked about "Jaccard's dissimilarity", please rethink if not one term is sufficient/justified!
Line 3329: Figure 11 - the legend (blue-red) must be explained explained like "(blue) low - dis/similarity - high (red)"
Line 287: again, a sentence is not yet a paragraph

Round 2

Reviewer 1 Report

Title:   Gathered wild food plants among diverse religious groups in Jhelum District, Punjab, Pakistan

Authors: M. Majeed et al.

The authors have made some changes, addressed  repetition, now include chutneys, put white lines between the photographs, explained the problem with the Venn diagrams and we are therefore almost there. However, one important change the authors have made, but one that was NOT asked for, should be reversed:  Figures 1 f, g, h are not the same as in the original and the authors are requested to use their original landscape/ecotype photographs! The new Figures 1 f, g, h do not refer to any of the study sites that the manuscript is based on.

Some minor corrections regarding spelling or grammar involve:

Line 60: ...this factor shapes...

Line 68: ....tried to articulate...

Line 77:  what does “these phenomena” refer to?

Line 312: delete “us”

Line 316: write “...in the immediate....”

Line 319: write “...of the bio-cultural...”

Line 323: “diseases”  (spelling)

I suggest the authors expand Line 200, in which they explain that some fruits are only collected in times of need and continue with  “Although not specifically addressed by us, taboos restricting the consumption of some plants and fruits under certain conditions have been described from many regions of the world, involving followers of various religions including Hindus.” The reference to be used would be Meyer-Rochow VB 2009 “Food taboos: origins and purposes” J Ethnobiol Ethnomed 5:18. Not mentioning that especially Hindus, but others possibly as well, have to follow specific rules in what they consume, especially when pregnant or menstruating, would leave readers of the article puzzled. For this reason it must be mentioned that there is the possibility, even if not examined in the research, that taboos may influence the uses of certain wild plans with regard to seasons or a consumer’s health condition, gender or age.  If not referred to on Line 200, another section could be 321-333 to mention the possible role of taboos in collecting and consuming wild plants. The authors in their response stated that “The recorded different taboos or temporary deviations were found more allied to medicinal uses than the food uses of the plant species”, but this is not entirely correct, especially when seen in connection with the Hindu.
